# Short-Term, Significant Gains from a 10-Day Field-Based Multi-Modal Outdoor Activity Camp with Time-Restricted Feeding Dissipate at Three-Month Follow-Up

**DOI:** 10.3390/jfmk10020229

**Published:** 2025-06-17

**Authors:** Katarina Milanović, Nikola Stojanović, Vladimir Miletić, Željko Rajković, Darko Stojanović, Vladimir Ilić, Milica Filipović, Slavka Durlević, Ana Orlić, Igor Ilić

**Affiliations:** 1Faculty of Sport and Physical Education, University of Belgrade, 11000 Belgrade, Serbia; katarinamanovski@yahoo.com (K.M.); vladimir.miletic@fsfv.bg.ac.rs (V.M.); zeljko.rajkovic@fsfv.bg.ac.rs (Ž.R.); vladimir.ilic@fsfv.bg.ac.rs (V.I.); ana.orlic@fsfv.bg.ac.rs (A.O.); 2Faculty of Sport and Physical Education, University of Niš, 18000 Niš, Serbia; 3Pedagogical Faculty in Vranje, University of Niš, 17000 Vranje, Serbia; darkos@pfvr.ni.ac.rs; 4Faculty of Sport and Physical Education, University of Priština in Kosovska Mitrovica, 38218 Leposavić, Serbia; milica.bojovic@pr.ac.rs (M.F.); igor.ilic@pr.ac.rs (I.I.); 5Faculty of Sports and Physical Education, University of Novi Sad, 21000 Novi Sad, Serbia; durlevicslavka3@gmail.com

**Keywords:** time-restricted feeding, outdoor mixed-modality exercise, body composition adaptations, physical activity level, mixed-effects modeling

## Abstract

**Objectives:** This single-arm, pre–post intervention study with a three-month follow-up aimed to determine whether a ten-day outdoor camp combining mixed-modality physical activity and time-restricted feeding elicits positive changes in multiple body-composition outcome measures and whether those changes persist at three-month follow-up. **Methods:** Forty healthy undergraduates (18 male, 22 female) participated in a 10-day outdoor camp that combined multi-modal physical activities (rock climbing and bouldering, swimming, hiking, applied paddling, survival skills, etc.) with a 13 h daily time-restricted feeding window. Body fat percentage, skeletal muscle percentage, body mass, total body-water percentage, visceral fat level, and skeletal muscle index were measured using the InBody 270 at baseline, immediately post-camp, and at the three-month follow-up. **Results:** Mixed-effects models with random intercepts for subject revealed significant reductions in body fat percentage (*β* = −1.63, *p* < 0.001) and visceral fat level (*β* = −0.72, *p* = 0.001), alongside increases in skeletal muscle percentage (*β* = 1.02, *p* < 0.001), skeletal muscle index (*β* = 0.30, *p* < 0.001), and total body-water percentage (*β* = 1.19, *p* < 0.001) from baseline to post-camp; no outcomes differed between baseline and follow-up and no time × sex interactions were observed. **Conclusions:** These findings indicate that a brief, intensive nature-based intervention can drive rapid, multidimensional improvements in body composition, but structured maintenance is required to sustain benefits.

## 1. Introduction

The rapid escalation of overweight and obesity rates represents a critical public health challenge, driven in part by sedentary lifestyles and ubiquitous access to energy-dense foods [1,2]. Excess adiposity and concomitant loss of lean body mass contribute to insulin resistance, dyslipidemia, and increased cardiovascular risk, thereby undermining functional capacity and quality of life. Traditional interventions, whether gym-based resistance training or home-based dietary counseling, often struggle with long-term adherence and ecological validity, as participants return to unsupervised environments where snacking, irregular meal timing, and inconsistent physical activity can erode early gains. Time-restricted feeding (TRF) has emerged as a promising dietary strategy, with early feeding windows of ~8–12 h demonstrating marked improvements in body weight, fat mass, and metabolic flexibility [3,4]. Importantly, when combined with structured exercise, the benefits of TRF appear to be amplified. Cui et al. [5] found that pairing TRF with resistance training yielded greater fat loss in overweight young adults than exercise alone, underscoring the synergy between meal timing and physical stimuli. This synergy asks what real-world intervention models can seamlessly integrate precise nutrition timing with diverse, moderate-intensity exercise stimuli to improve body composition?

Therefore, we argue that outdoor environments harness nature’s restorative and motivating properties, reducing stress, heightening enjoyment, and fostering sustained engagement in physical activity [6,7]. Outdoor resistance and body-weight exercises, such as those performed on park-based fitness structures, have yielded significant strength and body composition gains across various age groups. Leadbetter et al. [8] demonstrated that older adults achieved marked improvements in lower-body relative strength after a six-week outdoor training intervention, while Marcos-Pardo et al. [9] reported enhanced lean mass accrual and greater fat-mass reductions in middle-aged participants using outdoor fitness equipment. Similarly, Stanaszek et al. [10] demonstrated that wintertime outdoor activities led to reductions in body fat and waist circumference in adult men, accompanied by gains in strength and flexibility. Such findings suggest that natural settings offer varied neuromuscular and cardiovascular challenges, which can boost adherence through novelty and psychosocial enrichment. While these varied outdoor activities demonstrate the power of exercise in natural settings, equally rigorous control of dietary intake is necessary to fully understand how nutrition and movement interact to drive changes in body composition.

However, field-based interventions often lack rigorous dietary controls, yet precise meal timing might be a key determinant of metabolic adaptation. In isolated camps, meals can be fixed into daily windows of approximately 8–12 h, mirroring TRF protocols validated in laboratory studies [3,4]. Within this structured eating framework, the timing and composition of each meal should be calibrated to support the physical demands of the day. For instance, carbohydrate-focused breakfasts replenish glycogen stores and support cognitive performance, mid-day mixed-macronutrient meals facilitate recovery, and protein-emphasized evening meals enhance overnight muscle protein synthesis [11,12]. Czerwińska-Ledwig et al. [13] combined Nordic walking with TRF over 12 weeks and observed superior improvements in body composition, greater fat loss, and preservation of lean mass compared to exercise alone. Siedzik et al. [14] further illustrated that dietary composition (e.g., ketogenic fish-based diets) can drive rapid fat-mass declines without sacrificing strength, although they noted that minor adverse effects may arise. By synchronizing exercise and nutrition within the same daily window, camp protocols can maximize metabolic flexibility and compliance, yielding high-fidelity insights into body-composition dynamics.

Despite advances, few studies have simultaneously tracked multiple body-composition metrics, such as total mass, fat mass, skeletal muscle mass, hydration status, and visceral fat, within a single, tightly controlled intervention. Boughman et al. [15] observed that bioelectrical impedance measurements in backcountry environments can underreport fat mass relative to standard laboratory conditions. This highlights the necessity for rigorous, field-adapted protocols when assessing diverse compositional domains. Additionally, sex differences in response to both diet and exercise remain underexplored. Izzicupo et al. [16] found that male, but not female, children increased moderate-to-vigorous activity during a summer camp, while Laja García et al. [17] linked higher hydration status to lower fat mass and waist circumference in young adults, indicating potential sex-linked variance. Zhang et al. [2] documented a greater risk of visceral adiposity among female adolescents with low levels of outdoor activity. However, most studies either focus on single outcomes (e.g., fat loss or strength) or lack robust, longitudinal follow-ups to evaluate durability.

Building on these insights into the synergistic effects of immersive outdoor exercise and time-restricted feeding, the present study aims to elucidate how a high-variety, outdoor lifestyle regimen delivered in a fully provisioned natural setting with a consolidated feeding window modulates body composition, specifically adiposity, lean mass, hydration, and visceral fat. We hypothesized that immediately after the 10-day intervention, participants would exhibit significant reductions in body fat percentage and visceral fat levels, accompanied by increases in skeletal muscle mass and total body water. We further expect a sex-by-time interaction, with men demonstrating greater absolute lean-mass gains and women exhibiting proportionally larger relative decreases in adiposity. Finally, at the three-month follow-up, we anticipate that adiposity reductions will persist below baseline levels, while lean-mass and hydration gains will regress toward baseline in the absence of continued intervention.

## 2. Materials and Methods

### 2.1. Study Design and Procedures

A prospective, single-arm, within-subjects design evaluated body-composition changes over three time points: baseline, immediately post-intervention (final), and a three-month follow-up. Forty students from the Faculty of Sport enrolled in a 10-day outdoor practical camp, which included activities such as climbing, open-water swimming, windsurfing, paddling, hiking, and survival skills, participated. Baseline measurements were obtained using the Bioelectrical Impedance Analysis (BIA) device, the morning after overnight arrival, ensuring no prior physical exertion had occurred. Final measurements were collected the morning of day 11 (departure day), immediately after all activities were completed. The follow-up assessment was conducted three months later to evaluate the durability of any observed changes. All testing sessions were scheduled in the morning, and students were instructed to refrain from vigorous exercise for at least 24 h prior to each visit (see the flow chart in Figure 1).

### 2.2. Participants

A total of 40 healthy undergraduate students (18 males, 22 females) aged 22.5 ± 0.51 years from the Faculty of Sport and Physical Education at the University of Belgrade were enrolled. Male participants presented a mean height of 182.83 ± 7.05 cm and a mean body mass of 79.44 ± 8.11 kg, while female participants averaged 168.73 ± 5.36 cm in height and 61.17 ± 9.39 kg in body mass. Inclusion criteria comprised enrollment in the ten-day outdoor practical activities camp, absence of cardiovascular, metabolic, or musculoskeletal disorders, no use of medications affecting fluid or electrolyte balance, and refraining from vigorous exercise for at least 24 h before each assessment. Exclusion criteria included acute injury, recent illness, pregnancy, or declining to provide written informed consent. The study protocol received approval from the Faculty of Sport and Physical Education Ethics Committee at the University of Belgrade (approval number: 2-1295/22-2; date of approval: 28 June 2022), and all procedures adhered to the Declaration of Helsinki.

### 2.3. Intervention Protocol and Internal Load Monitoring

Over the ten-day outdoor practical camp, three cohorts of 13–14 students each completed the same technical and conditioning modules, differing only in the daily sequence of activities to accommodate transport and site availability. Ethical approval was obtained from the University’s Institutional Review Board, and all participants provided written informed consent before data collection. Each morning began with a standardized 30 min dynamic mobilization and stretching routine led by certified instructors (instructor-to-student ratio ≈ 1:6), ensuring that all students were familiar with proper warm-up mechanics and injury-prevention cues. Before the first session, participants attended a brief orientation on the Borg 6–20 RPE scale [18], which included practice ratings during a sample drill to ensure the reliability of subsequent self-reports. Practical days then unfolded around three 90 min blocks: two sessions devoted entirely to motor-skill and technical drill work (e.g., bouldering techniques, windsurfing, applied paddling), and one hybrid block split equally between skill refinement and vigorous conditioning exercises (see Table 1). Environmental conditions (air temperature and relative humidity) were logged daily to contextualize exertion ratings. All practical sites were equipped with first-aid stations and emergency communication to ensure participant safety.

After each 90 min block, students recorded their perceived exertion using paper log sheets. The RPEs from these three sessions were averaged to yield a single daily RPE score for each student. Attendance exceeded 95% across all sessions, and compliance with RPE reporting was effectively 100%, thanks to post-session reminders and on-site form collection. To control for potential order effects, the sequence of modules was counterbalanced across groups using a Latin-square design, ensuring that no single activity consistently fell in an early or late session slot.

### 2.4. Dietary Intake and Meal Timing

Participants consumed three standardized meals daily to meet the energy demands of sustained outdoor training while ensuring safety in the natural environment. Portion sizes for breakfast, lunch, and dinner were individually calibrated to each student’s total daily energy expenditure, calculated via the Mifflin–St. Jeor equation [19] with an appropriate activity factor [20]. As detailed in Table 2, the 24 h camp schedule, which included three 90 min practical sessions, morning gymnastics, and evening activities, yielded an average physical activity level (PAL) of 2.52. This PAL was used to scale each participant’s resting energy expenditure (REE) to their estimated daily energy needs. Breakfast at 08:00 comprised locally available shelf-stable canned porridge (oat-based packs), fruit compote, and whole-grain crackers, delivering approximately 60% of energy from carbohydrates, 20% from protein, and 20% from fat—consistent with guidelines that emphasize morning carbohydrate intake to restore glycogen and support cognitive function [12]. A freshly prepared lunch provided a balanced distribution of 55% carbohydrate, 25% protein, and 20% fat to replenish substrates between the morning and afternoon practical sessions. Dinner at 19:00 featured canned meat-and-bean stews and canned fish varieties common in Serbian markets, supplying roughly 35% of energy from protein, 30% from carbohydrates, and 35% from fat—a macronutrient profile shown to enhance overnight muscle protein synthesis and support recovery [11]. The total daily protein intake averaged approximately 1.5 g/kg body weight to ensure adequate substrate availability for anabolism and tissue repair. No food was permitted after 23:00 to minimize wildlife encounters, and the camp remained strictly food-free throughout the night.

### 2.5. Measurements

Baseline assessments were performed between 07:15 and 08:45 in a temperature-controlled laboratory before breakfast. Morning exercises and camp cleanup were omitted on the testing days. Stature was measured to the nearest 0.1 cm using a wall-mounted stadiometer with sliding headboard from a Swiss anthropometry set (GPM Anthropological Instruments, DKSH Switzerland Ltd., Zurich, Switzerland). Body mass and composition were assessed using a single, direct-segmental, multi-frequency bioelectrical impedance analyzer (InBody 270; InBody Co., Seoul, Republic of Korea), auto-calibrated each test day. The InBody 270 has demonstrated excellent validity against dual-energy X-ray absorptiometry [21,22,23,24]. The InBody 270 uses eight touchpoints, two per each hand and foot, to perform 10 impedance measurements by assessing five body segments (right arm, left arm, trunk, right leg, and left leg) using two different frequencies: 20 kHz and 100 kHz [25]. Together with the user-entered standing height, sex and age, along with the auto-calculated body fat mass (BFM), the firmware applies the standard BIA sequence: segmental total body water (TBW) from the impedance index, fat-free mass (FFM) from total body water (TBW/0.73), skeletal muscle mass (SMM) via proprietary regression on segmental FFM, and body fat mass (BFM) by subtraction [26]. From the BIA output, we extracted the following: total body mass (BM, kg), body fat mass (BFM, kg), skeletal muscle mass (SMM, kg), total body water (TBW, kg), and visceral fat level (VFL, arbitrary units). We then derived the following additional variables: body fat percentage (%BFM = [BFM/BM] × 100), skeletal muscle mass percentage (%SMM = [SMM/TT] × 100), total body water percentage (%TBW = [TBW/TT] × 100), and the skeletal muscle index (SMI), calculated as SMM divided by stature squared [27]. Participants adhered to the manufacturer-recommended pre-test controls: adequate hydration, no alcohol or excess caffeine for 24 h, no strenuous exercise for at least eight hours, an overnight fast with the final meal consumed at least four hours before testing, and no creams, lotions, or ointments applied to the hands or feet. On arrival, they voided their bladder, removed all jewelry and metallic objects, and changed into underwear. After standing for seven minutes to stabilize fluid distribution, palms and soles were wiped with 70% isopropyl alcohol and air-dried. Two certified exercise physiologists entered each participant’s ID, height, sex, and age into the analyzer and delivered standardized, step-by-step instructions on posture, electrode contact, and breathing, guiding each participant through the entire measurement process. Participants stood barefoot on the foot electrodes with their heels aligned to the rear guides and gripped the hand electrodes with thumbs and fingers placed on the designated sensors, maintaining arms abducted ~15° from the trunk and legs slightly apart. The BIA’s voice prompts and staff guidance ensured participants’ immobility during impedance recording.

### 2.6. Statistical Analyses

All statistical analyses were conducted in RStudio (version 2024.12.1+563; Posit PBC, Boston, MA, USA). Descriptive statistics (means ± SD) were calculated for each BIA measure—body fat mass (BFM), skeletal muscle mass (SMM), total body mass (BM), total body water (TBW), visceral fat level (VFL), and skeletal muscle index (SMI)—stratified by sex and time point (baseline, final, three-month follow-up). Model assumptions for the inferential analyses were evaluated using simulation-based diagnostics from the DHARMa package (version 0.4.7) [28]. One thousand scaled residuals per model were generated to test for uniformity, dispersion, and outliers. Inferential tests focused on six variables chosen to represent adiposity (body fat percentage and VFL), lean mass (skeletal muscle percentage and height-adjusted SMI), hydration status (percent total body water, or TBW), and overall body mass (body mass, or BM). REML-fitted linear mixed-effects models were used with lmerTest (version 3.1-3) [29], which included fixed effects for time, sex, and their interaction, as well as a random intercept for subject. Apart from interaction effects, simple-effect contrasts (final vs. baseline, three-month follow-up vs. baseline, final vs. three-month follow-up) were estimated via emmeans (version 1.11.1) [30]. Cohen’s *d* effect sizes were computed by dividing contrast estimates by the model residual SD. *p*-values were adjusted using the Benjamini–Hochberg procedure. Statistical significance was set at *p* < 0.05.

## 3. Results

Table 3 presents descriptive statistics for all absolute BIA measures, such as body mass, skeletal muscle mass, body fat mass, total body water, visceral fat level, and skeletal muscle index, as means and standard deviations. The sample comprised 40 participants (18 male, 22 female), each assessed at baseline, final, and three-month follow-up (120 observations per variable); measures exhibited moderate variability but no extreme values. The mean estimated daily energy expenditure was 4615 ± 285 kcal for male participants and 3507 ± 298 kcal for female participants. When aggregated over ten days and across all participants, the mean daily RPE was 11.89 (SD = 0.70), corresponding to a light-to-moderate exertion level according to Borg’s original scale interpretation.

Before interpreting the mixed-model estimates, we evaluated model assumptions using simulation-based residual diagnostics. For each outcome, the scaled residuals did not deviate from the expected uniform distribution (KSD = 0.072–0.119, *p* = 0.068–0.557), and nonparametric tests of dispersion confirmed neither over- nor under-dispersion (dispersion index = 0.969–0.971, *p* = 0.936–0.944). Outlier tests likewise showed extreme residuals at or below the nominal 0.2% rate (≤0.83%, *p* = 0.213). Together, these diagnostics indicate that the key assumptions of normality, homoscedasticity, the absence of outliers, and the absence of undue leverage are met, supporting the validity of the inferential results reported below.

Following model diagnostics, mixed-effects models (REML; random intercepts for subject) were fitted to the BIA measures as follows: body fat percentage, skeletal muscle percentage, body mass (kg), total body-water percentage, visceral fat level, and skeletal muscle index (kg/m^2^)—to assess changes over three time points (baseline, final, three-month follow-up) and between sexes. All *β*-coefficients reported below are unstandardized estimates. Simple-effect contrasts (final vs. baseline, three-month follow-up vs. baseline, and final vs. three-month follow-up) are presented graphically, along with corresponding effect sizes and *p*-values (see Figure 2). Body fat percentage reduced significantly from baseline to final (*β* = −1.63, *t*(76) = −3.79, *p* < 0.001), whereas the three-month follow-up condition did not differ from baseline (*p* = 0.254); female participants carried higher fat percentages than males (*β* = 10.86, *t*(41.3) = 6.58, *p* < 0.001). Skeletal muscle percentage increased from baseline to final (*β* = 1.02, *t*(76) = 4.16, *p* < 0.001) but showed no change between final and three-month follow-up (*p* = 0.149); females had lower muscle percentages than males (*β* = −7.86, *t*(41.4) = −8.46, *p* < 0.001). Body mass remained stable across all comparisons (final: *β* = 0.14, *t*(76) = 0.35, *p* = 0.730; three-month follow-up: *p* = 0.740), although female subjects weighed less than males (*β* = −18.31, *t*(39) = −6.41, *p* < 0.001). Total body-water percentage rose significantly from baseline to final (*β* = 1.19, *t*(76) = 3.84, *p* < 0.001) but did not differ between final and three-month follow-up (*p* = 0.085); females exhibited lower hydration percentages than males (*β* = −7.99, *t*(41.1) = −6.53, *p* < 0.001). Visceral fat level decreased from baseline to final (*β* = −0.72, *t*(76) = −3.51, *p* < 0.001) with no change between final and three-month follow-up (*p* = 0.283), where females had significantly higher visceral fat levels than males (*β* = 2.36, *t*(41.9) = 3.24, *p* = 0.002). Finally, skeletal muscle index increased from baseline to final (*β* = 0.30, *t*(76) = 4.75, *p* < 0.001) but not between final and three-month follow-up (*p* = 0.569), and female participants had lower SMI than males (*β* = −2.93, *t*(40.5) = −10.73, *p* < 0.001). No time × sex interactions were significant, indicating parallel changes across sexes.

In addition, we observed significant lean-mass gains in the left leg (*β* = 0.350, *t*(76) = 4.80, *p* < 0.001) and right leg (*β* = 0.32, *t*(76) = 4.06, *p* < 0.001); these limb-specific data are not shown in Table 3 or Figure 2 for brevity. Overall, the 10-day intervention resulted in consistent reductions in adiposity and visceral fat, accompanied by concomitant increases in lean mass and hydration, with parallel effects observed in both men and women.

## 4. Discussion

The present ten-day outdoor camp, which integrates time-restricted feeding (TRF) with a varied, nature-based physical activity regimen, produced rapid and meaningful adaptations in multiple body composition domains. Specifically, participants experienced significant reductions in body fat percentage and visceral fat, while simultaneously increasing their skeletal muscle percentage and skeletal muscle index, and enhancing their total body water percentage. These changes were evident immediately post-intervention, but by the three-month follow-up assessment, values had reverted to baseline levels, indicating that the camp-induced improvements were not sustained in an unsupervised environment. Moreover, no significant time × sex interactions were observed, indicating parallel responses in male and female undergraduates, despite initial sex differences in adiposity and lean mass.

Our findings align with a growing body of work demonstrating that TRF can facilitate fat loss within remarkably short time frames when combined with moderate to vigorous physical activity. Early TRF trials in controlled laboratory settings, employing 8–12 h feeding windows, have consistently demonstrated 0.4–0.6% reductions in fat mass over four weeks, alongside improvements in insulin sensitivity and lipid profiles [3,4]. In contrast, our cohort achieved a somewhat greater 1.6% decrease in body fat percentage in just ten days. This accelerated lipolysis likely reflects the elevated energy deficit created by an estimated average PAL of 2.52, which exceeds the level of very physically active [20]. Scaling each participant’s REE using the Mifflin–St. Jeor equation [19] and multiplying by the empirically derived PAL (Table 2), we ensured that meal portions precisely matched the high caloric demands of three daily 90 min practical sessions, morning gymnastics, and evening programming.

The diverse, formative activity program might have recruited both slow- and fast-twitch muscle fibers through skill drills (e.g., rock climbing, paddling) and vigorous conditioning (e.g., hybrid class 3 sessions), driving neuromuscular adaptations that manifested as increases in skeletal muscle percentage (*β* ≈ 1.02, *p* < 0.001) and skeletal muscle index (*β* ≈ 0.30, *p* < 0.001). While traditional resistance-training interventions often require several weeks to elicit similar hypertrophic responses [8,9], the camp’s immersive, multi-modal approach appears to accelerate protein-synthesis signaling pathways, perhaps via repeated high-intensity eccentric loads inherent to natural obstacles. However, although the reported mean daily RPE was relatively low (RPE ≈ 11, corresponding to “light to somewhat hard” effort), several factors may explain the observed muscle gain beyond what the average RPE alone suggests. First, one daily Class 3 session (vigorous effort; see Table 2) would have provided a muscle-damaging, eccentric overload stimulus, an established driver of myofibrillar repair and hypertrophy, even when embedded in an otherwise moderate-intensity framework [31]. Second, the novelty and variety of multi-modal activities such as rock climbing, which requires sustained isometric contractions, hiking over inclined terrain, imposing prolonged time under tension (TUT), and paddling, which involves both concentric and eccentric muscle actions, likely produced cumulative mechanical and metabolic stress not fully captured by a single daily average RPE value. Specifically, prolonged TUT is known to contribute to hypertrophic signaling, especially when paired with eccentric loading. However, large eccentric overloads are not strictly necessary to induce muscle growth. To be more precise, when total training volume is equated, TUT alone in low-load, varied-activity contexts can still effectively stimulate muscle protein synthesis [31].

Furthermore, we provided individualized protein-feeding protocols in line with recommendations for strength-trained individuals (1.5 g/kg/day), recognizing that resistance exercise necessitates protein intake exceeding the Recommended Daily Allowance (RDA) to support early-phase hypertrophic gains [32]. Nonetheless, uncontrolled factors, such as additional unstructured movement on steep camp terrain (traversing to camp facilities, carrying equipment up inclines), may have influenced net anabolism. Notably, despite the overall low-to-moderate average RPE, both structured and unstructured activities likely converged to drive leg-specific hypertrophy. For instance, the single high-intensity Class 3 session (rigorous uphill hiking with heavy packs) would have provided a muscle-damaging eccentric stimulus to the lower limbs. On the other hand, continuous low-level loading, such as walking on sloped paths, moving gear between stations, and performing camp chores on uneven and steep ground, would have increased TUT for leg musculature throughout each day. This combination of intermittent vigorous overload and prolonged daily loading might underpin the observed significant gains in lean mass for the legs (left leg: *β* = 0.35, *p* < 0.001; right leg: *β* = 0.3, *p* < 0.001). In contrast, the arms and trunk showed no significant hypertrophic changes, suggesting that the camp’s predominantly lower-body, incline-driven demands could be the primary drivers of regional muscle growth. It is also plausible that some participants experienced caloric deficits despite estimated daily caloric requirements, thereby amplifying fat mass loss but still yielding modest skeletal muscle gains through enhanced protein utilization and neuromuscular adaptation [32]. Therefore, the combination of a single high-intensity session might elevate TUT from varied activities, adequate protein intake, and structured and unstructured everyday movement, likely working synergistically to produce rapid hypertrophic responses despite a relatively low average daily RPE. In addition, concomitant expansions in total body water percentage (*β* ≈ 1.19, *p* < 0.001) may suggest glycogen repletion and plasma volume increases, both hallmarks of effective training stimuli [33]. Importantly, the visceral fat level decreased significantly from baseline to the final measurement (*β* ≈ −0.72, *p* < 0.001). These central adiposity reductions likely stem from synergistic mechanisms: TRF-mediated enhancements in overnight lipolysis [34], exercise-induced upregulation of catecholamine-driven fat oxidation [35], and the stress-buffering effects of nature exposure, which can attenuate cortisol spikes that promote abdominal fat storage [6,7].

A pivotal and somewhat unexpected finding was the complete return of all measured body-composition variables to baseline levels at the three-month follow-up. This reversion emphasizes the importance of ongoing stimulation and dietary monitoring for maintaining health. The literature on detraining demonstrates that gains in muscle strength and hypertrophy can dissipate rapidly once structured training ceases. Meta-analytic evidence suggests that even brief periods (4–6 weeks) of inactivity lead to substantial reductions in muscle size and strength [36,37]. Similarly, weight-loss interventions commonly experience high recidivism: only about 25% of individuals maintain reduced body weight over the long term without ongoing support [38]. In our free-living context, participants returned to unsupervised routines, lacking the camp’s precise meal timing and structured activities, which permitted their habitual dietary and activity patterns to reassert themselves. Behavioral theories further elucidate this phenomenon. The transtheoretical model posits that maintenance of health behaviors requires distinct self-regulatory skills beyond those used during active change [39]. These authors argue that individuals may revert to prior habits without structured reinforcement, such as supervised exercise or scheduled TRF windows. Our findings thus highlight that, while short-term, intensive interventions can rapidly alter physiology, embedding those changes into sustainable lifestyle practices is essential for achieving enduring outcomes.

The observed rapid adaptations likely reflect acute physiological processes. TRF aligns feeding with circadian clocks in peripheral tissues, thereby optimizing mitochondrial function and enhancing fatty acid oxidation during fasting periods [40,41]. Exercise in natural settings may further potentiate these effects: green exercise research shows that outdoor environments amplify parasympathetic activation and attenuate stress hormone release, thereby improving energy balance regulation [42,43]. The combined TRF-exercise stimulus also plausibly enhances autophagy and mitochondrial biogenesis, processes crucial to metabolic health and body composition remodeling [44]. However, suboptimal dietary adherence during camp, particularly under-consumption in afternoon and evening sessions, may have compounded the energy deficit, accentuating fat loss at the expense of maximal muscle protein synthesis. Future protocols should incorporate easily consumable, nutrient-dense supplements (whey protein shakes) within the TRF window to ensure adequate amino acid availability [11]. Moreover, integrating remote monitoring (e.g., continuous glucose monitoring, activity trackers) and digital nudges could support compliance and extend the benefits of camp beyond the ten days.

Despite its strengths, the study has limitations. The single-arm design prevents causal attribution; a randomized controlled trial with an active comparison (gym-based TRF or non-TRF camp) is needed to isolate the unique contributions of each component. Although the BIA provides practical, multi-frequency bioelectrical impedance analysis, it is less precise than dual-energy X-ray absorptiometry or MRI for quantifying regional adiposity and lean tissue [21,22,23,24,27]. Our homogeneous sample of healthy, sport-science undergraduates limits generalizability to other age groups, clinical populations, or athletic populations. Despite high compliance, self-reported RPE and meal adherence logs remain subject to bias. Finally, the lack of direct mechanistic measures, such as substrate oxidation rates, hormonal assays, or muscle biopsy data, renders our mechanistic explanations inferential rather than definitive.

To build on these findings, future studies should employ randomized designs comparing camp-based TRF/exercise interventions with traditional gym- or home-based programs, using gold-standard compositional imaging (DXA) and metabolic monitoring (e.g., indirect calorimetry, continuous glucose monitoring). Extending follow-up to six months or longer, with intermittent “booster” camps or digital coaching, can elucidate maintenance strategies. Incorporating assessments of circadian hormone rhythms, gene-expression profiling in muscle and adipose tissue, and psychological mediators (e.g., nature-relatedness, motivation) will deepen mechanistic insight. Finally, investigating sex-specific endocrine responses, particularly in female participants, may reveal tailored protocols that optimize outcomes based on sex [16,17].

In summary, our findings confirmed that the 10-day camp, which combined TRF and outdoor exercise, elicited significant reductions in body fat percentage and visceral fat, along with increases in skeletal muscle mass and total body water percentage, immediately after the camp. However, reductions in adiposity did not persist at the three-month follow-up, gains in lean mass and hydration regressed toward baseline, and we did not observe the expected sex-by-time interaction at either post-camp or the three-month follow-up.

## 5. Conclusions

Integrating time-restricted feeding with a high-variety, nature-immersive activity regimen yields rapid and robust improvements in body composition and hydration within ten days. Although both men and women benefited, we did not observe the predicted sex-by-time interactions. However, these gains dissipate without structured maintenance, underscoring the importance of sustained behavioral and environmental support. By harnessing the synergistic effects of circadian-aligned nutrition, heterogeneous mechanical stimuli, and restorative natural settings, this multi-modal camp provides a potent and scalable blueprint for combating obesity. The challenge moving forward is translating these short-term physiological shifts into long-term, sustainable lifestyle changes.

## Figures and Tables

**Figure 1 jfmk-10-00229-f001:**
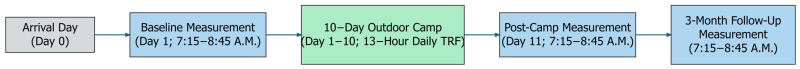
Flowchart of study timeline and intervention activities.

**Figure 2 jfmk-10-00229-f002:**
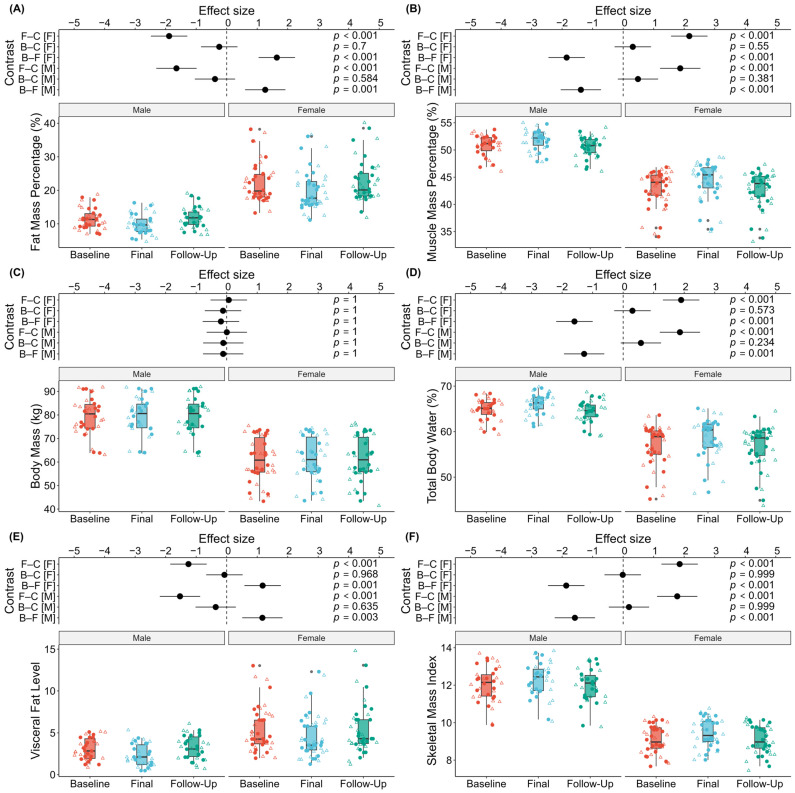
Effect sizes and distributions of observed versus model-predicted BIA outcomes across time and sex. Panels A–F correspond to (**A**) body fat percentage, (**B**) skeletal muscle percentage, (**C**) body mass (kg), (**D**) total body-water percentage, (**E**) visceral fat level, and (**F**) skeletal muscle index (kg/m^2^). In each panel, the upper subpanel displays Cohen’s d for the pairwise contrasts final versus baseline (F–B), three-month follow-up versus baseline (C–B), and final versus three-month follow-up (F–C) for females [F] and males [M], with 95% confidence intervals and *p*-values annotated. The lower subpanel displays boxplots of the model-predicted values (boxes represent the median ± interquartile range; whiskers represent 1.5× the interquartile range) at each time point on the x-axis, overlaid with jittered points. Circles denote observed values, and triangles denote predicted values for individual subjects. Color coding indicates time point (baseline = red, final = blue, three-month follow-up = green). Panels are arranged in a 2 × 3 grid with males on the left and females on the right.

**Table 1 jfmk-10-00229-t001:** Daily practical activity schedule.

Day	Session 1 (90 min)	Session 2 (90 min)	Session 3 (90 min)
Day 1	Rock Climbing and Bouldering	Nautical Operations	Environmental Stewardship
Day 2	Open-Water Swimming	Rock Climbing and Bouldering	Nautical Operations
Day 3	Windsurfing	Open-Water Swimming	Rock Climbing and Bouldering
Day 4	Applied Paddling	Windsurfing	Open-Water Swimming
Day 5	Hiking and Navigation	Applied Paddling	Windsurfing
Day 6	Obstacle Course Movement	Hiking and Navigation	Applied Paddling
Day 7	Campcraft Fundamentals	Obstacle Course Movement	Hiking and Navigation
Day 8	Survival Skills	Campcraft Fundamentals	Obstacle Course Movement
Day 9	Rafting Expedition	Survival Skills	Campcraft Fundamentals
Day 10	Environmental Stewardship	Rafting Expedition	Survival Skills

**Table 2 jfmk-10-00229-t002:** Daily schedule and PAL calculation.

Time	Activity	Duration (h)	REE ×	Weighted REE (h)
23:00–07:00	Sleep/night duty	8.0	1.0	8.0
07:00–07:30	Wake-up and toilet	0.5	1.0	0.5
07:30–08:00	Morning gymnastics	0.5	5.0	2.5
08:00–09:00	Camp cleanup and breakfast	1.0	2.5	2.5
09:00–10:30	Class 1 (practical PA)	1.5	5.0	7.5
10:30–11:00	Break (moving around)	0.5	2.5	1.25
11:00–12:30	Class 2 (practical PA)	1.5	5.0	7.5
12:30–14:00	Free time (light activities)	1.5	2.5	3.75
14:00–16:00	Lunch and light free time	2.0	2.5	5.0
16:00–17:30	Class 3 (vigorous PA, RPE > 13)	1.5	7.0	10.5
17:30–19:00	Lecture/presentations (sitting)	1.5	1.0	1.5
19:00–21:00	Dinner and prep (light)	2.0	2.5	5.0
21:00–23:00	Evening program (light)	2.0	2.5	5.0
TOTAL		24.0		60.5
PAL =				2.52

Note. Duration reflects hours spent in each activity; “REE ×” indicates the multiple of resting energy expenditure applied to that activity; “Weighted REE (h)” equals Duration × REE ×; TOTAL is the sum of weighted REE-hours over 24 h; PAL (physical activity level) is TOTAL ÷ 24.

**Table 3 jfmk-10-00229-t003:** Descriptive statistics and percentage changes in BIA-derived body composition measures by sex across baseline, final, and three-month follow-up assessments.

Variable	Sex	Baseline	Final	Follow-Up	ΔF–B (%)	ΔC–B (%)	ΔF–C (%)
BFM (kg)	male	9.06 ± 2.59	7.79 ± 2.57	9.56 ± 3.12	−13.9%	5.7%	−17.0%
	female	13.99 ± 5.49	12.72 ± 5.40	14.28 ± 6.03	−9.6%	2.0%	−10.6%
SMI	male	12.02 ± 0.98	12.31 ± 0.98	11.98 ± 0.89	2.5%	−0.2%	2.8%
	female	9.09 ± 0.74	9.44 ± 0.80	9.09 ± 0.79	3.8%	0.0%	3.9%
SMM (kg)	male	40.36 ± 4.88	41.22 ± 4.83	40.09 ± 4.46	2.2%	−0.5%	2.8%
	female	26.03 ± 3.42	26.98 ± 3.61	25.93 ± 3.43	3.6%	−0.4%	4.1%
TBW (kg)	male	51.53 ± 5.84	52.56 ± 5.84	51.13 ± 5.39	2.0%	−0.7%	2.8%
	female	34.47 ± 4.21	35.53 ± 4.44	34.34 ± 4.21	3.1%	−0.4%	3.5%
BM (kg)	male	79.35 ± 8.21	79.49 ± 8.31	79.49 ± 8.29	0.2%	0.2%	0.0%
	female	61.04 ± 9.45	61.28 ± 9.34	61.19 ± 9.82	0.4%	0.2%	0.3%
VFL	male	3.00 ± 1.33	2.28 ± 1.36	3.22 ± 1.48	−24.9%	9.1%	−27.2%
	female	5.36 ± 2.66	4.64 ± 2.61	5.41 ± 3.17	−14.5%	−0.7%	−9.3%

Note. BFM = body fat mass (kg); SMI = skeletal muscle index (kg/m^2^); SMM = skeletal muscle mass (kg); TBW = total body-water (kg); BM = body mass (kg); VFL = visceral fat level (unitless index). ΔF–B (%) = percent change from baseline to final; ΔC–B (%) = percent change from baseline to three-month follow-up; ΔF–C (%) = percent change from three-month follow-up to final. Values are presented as mean ± SD (descriptives) or mean Δ% (percent-change columns).

## Data Availability

The datasets generated and/or analyzed during the current study are available from the corresponding author upon reasonable request.

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
