# Peer review of "Short-Term, Significant Gains from a 10-Day Field-Based Multi-Modal Outdoor Activity Camp with Time-Restricted Feeding Dissipate at Three-Month Follow-Up"

_jfmk, 2025, doi:10.3390/jfmk10020229_

Round 1
Reviewer 1 Report
Comments and Suggestions for Authors
This study aims to examine the effects of a short-term camp combining time-restricted feeding (TRF) and varied outdoor physical activity on several body composition indicators, also considering potential sex differences.
The introduction provides a clear contextualization of the problem and presents relevant studies supporting the effectiveness of TRF and outdoor exercise. The authors also identify a gap in the literature, as most previous studies do not adequately control dietary intake or measure outcomes using diverse body composition metrics. The objective of the study is clearly stated at the end of the introduction; however, the hypotheses are not explicitly formulated, and I suggest they be included. I would also recommend summarizing the introduction to avoid redundancy in the definitions of concepts such as TRF and outdoor exercise.
Regarding the design and methodology, the study is supported by a solid and well-described methodological framework, which enhances its replicability. The control of diet and the structured physical activity protocol are key strengths of this research. However, despite employing a within-subjects longitudinal design, there is no control group with either no intervention or an alternative intervention, which limits the causal inference of the results. Another limitation is the use of a young and physically active sample, which may restrict the generalizability of the findings.
In terms of the results, the authors present them in a structured and comprehensive manner, starting with descriptive statistics and then moving on to mixed-model results, including assumption checks and the effects of time, sex, and their interaction. The contrasts are appropriate, and the figures are clear. I have two minor suggestions: first, the beta coefficients are not always clearly stated as standardized or unstandardized, which complicates interpretation; second, there is no brief summary at the end of the results section—I recommend including at least one sentence that synthesizes the main findings and sets the stage for the discussion.
The discussion is well developed and well-founded. The authors acknowledge the study's limitations and propose future research directions. However, I recommend closing with a more concise summary and revisiting the hypotheses (once they are clearly stated in the introduction).
Author Response
All responses are listed in a point-by-point format for reviewers. Please see the attachment!

Reviewer 2 Report
Comments and Suggestions for Authors
This manuscript presents a novel field-based study investigating the impact of a multi-modal outdoor activity camp combined with time-restricted feeding on body composition in healthy young adults. The practical relevance of exploring such an engaging, real-world intervention is high, and the authors have clearly invested significant effort in designing and executing the 10-day program. The finding that initial positive changes in body composition were not sustained at the 3-month follow-up is a crucial and honestly reported result, offering valuable insights into the challenges of long-term adherence without ongoing support. The detailed description of the intervention protocol and the appropriate statistical analyses are also strengths of this work.
Specific Comments and Recommendations:
Throughout the manuscript, "InBody" (a product name) should be replaced with the general scientific term "Bioelectrical Impedance Analysis (BIA)" when you mention about the assessment."
Title: TRF in the title may not be immediately clear to all readers. It is recommended to spell out "Time-Restricted Feeding" in the title for broader accessibility.
Title: The current title, "Short-term, big results," while catchy, does not fully reflect the key finding that the observed effects disappeared after three months. A more realistic title that encapsulates both the initial impact and the lack of sustainability would be more appropriate.
Abstract: The abstract should briefly but clearly describe the core components of the intervention (e.g., "a 10-day outdoor camp featuring multi-modal physical activities [climbing, swimming, hiking etc.] combined with an X-hour time-restricted feeding window").
Abstract: It is crucial to explicitly state in the Abstract that this was a "single-arm, pre-post intervention study with a 3-month follow-up" to accurately set reader expectations regarding the study's design and limitations in inferring causality.
Discussion: The rationale provided for the observed decrease in body fat mass is plausible. However, the explanation for the increase in skeletal muscle mass warrants further elaboration. The reported mean daily RPE of 11 is relatively low. Please expand the discussion on muscle gain. Consider factors such as protein intake and specific activity elicit muscle hypertrophy, or initial muscle gain for student athletes. Was protein intake optimized and sufficient to support anabolism under these conditions? While the average RPE is low-to-moderate, were there specific high-intensity, muscle-damaging components (e.g., eccentric loading during hiking, climbing) that might have provided a sufficient stimulus for hypertrophy despite the overall RPE? Could the variety and novelty of multi-modal activities have played a role not fully captured by a single daily RPE average?
Given that BIA results are a main focus, and impedance measurements can be sensitive to various factors, more detailed methodological information is essential to assure readers of the reliability and control of potential biases. Please provide more comprehensive details in the Measurements section. Was a single BIA device used for all measurements, or multiple? Standardized posture and duration of rest (e.g., supine rest for X minutes)? Removal of metallic objects/jewelry? What were participants instructed to wear? How was consistent and proper contact with hand and foot electrodes ensured (e.g., cleaning of electrodes, participant instruction on grip/stance)?
To the extent publicly available or citable from the manufacturer (InBody 270), please describe the algorithm. What specific variables are input by the device (e.g., height, weight, age, sex, impedance at various frequencies)? Are there any specific equations or principles used by this model that are citable?
For each derived BIA index reported (SMM, SMI, VFL etc.), please cite previous research (if available, specifically for the InBody 270 or similar multi-frequency segmental BIA devices) that has validated these specific parameters against gold-standard methods (e.g., DXA, MRI) in comparable populations.
Figure 1: This visual style differs from other figures/tables. Consider revising for better visual consistency. The middle layer ("10-Day Outdoor Practical Course" activities) largely duplicates information in Table 1. Figure 1 would be more impactful if it focused on aspects not detailed in Tables 1 and 2, specifically a clear timeline illustrating: Timing of each measurement point (Baseline, Final, Follow-up), Key intervention components like the daily TRF window (e.g., "XX-hour feeding window daily").
Table 1: Please enhance Table 1 by including the timing of key logistical and measurement events, such as: Arrival (Day -1 or Day 0). Baseline measurement timing (e.g., "Morning of Day 1, pre-activity"). Final measurement timing (e.g., "Morning of Day 11, post-activity"). Departure (Day 11). This would provide a more complete timeline within the activity schedule.
Table 3: The term "Control" for the 3-month follow-up assessment is potentially confusing. "Control" usually implies a control group or a pre-intervention baseline state. Since this is a post-intervention follow-up, please change "Control" to "3-Month Follow-Up" (or simply "Follow-Up") for consistency with the main text and to avoid misinterpretation.
L155-156: The mean daily RPE is a result of the intervention. This information should be moved to the "Results" section.
L163-165: The resulting average caloric intake should be presented in the Result.
Author Response

(The authors gave the same response as above.)

Round 2
Reviewer 2 Report
Comments and Suggestions for Authors
The revised version is much improved, and I have no further comments for correction.
Author Response
Dear Reviewer,
Thank you very much for your kind words and for guiding us through the revision process. We're glad the revised version meets your expectations.
Best regards!